# Mycotoxin Exposure in Children through Breakfast Cereal Consumption in Chile

**DOI:** 10.3390/toxins14050324

**Published:** 2022-05-03

**Authors:** Claudia Foerster, Liliam Monsalve, Gisela Ríos-Gajardo

**Affiliations:** 1Institute of Agri-Food, Animal and Environmental Sciences (ICA3), Universidad de O’Higgins, San Fernando 3070000, Chile; liliam.monsalve@uoh.cl; 2Department of Food Science and Technology, Faculty of Pharmacy, Universidad de Concepción, Concepción 4030000, Chile; grios@udec.cl

**Keywords:** risk assessment, breakfast cereals, aflatoxins, ochratoxin A, zearalenone, fumonisins, deoxynivalenol, T-2/HT-2 toxins, children

## Abstract

Mycotoxins are unavoidable contaminants produced by fungi in food, especially grains. This study aimed to measure the occurrence and levels of total aflatoxins (AFs); ochratoxin A (OTA); zearalenone (ZEN); fumonisins B1, B2, and B3 (FUM); deoxynivalenol (DON); and T-2/HT-2 toxins in the four most commonly consumed breakfast cereals in Chile and to assess mycotoxin exposure and risk in children aged 2 to 13 years due to cereal consumption. In this study, a total of 110 batches with three subsamples of the four brands were sampled in supermarkets from November 2019 to June 2021. Samples were analyzed by Veratox^®^ ELISA (Neogen). Exposure was assessed by estimated daily intake (EDI) considering the levels found in a modified lower bound (mLB) and upper bound (UB). Risk was estimated by margin of exposure (MOE) in the case of OTA and AFs and hazard quotient (HQ) for the rest of the mycotoxins. No T2/HT2 toxins were detected. Few samples had quantifiable levels of ZEN, FUM, and DON except for brand 1, with a mean (standard deviation, SD) of 54 (20), 1552 (351), and 706 (218) ng/g, respectively. In addition, three FUM samples and one DON sample had values over the Chilean regulation. Brands 2, 3, and 4 had quantifiable levels of AFs, with mean (SD) values of 1.3 (0.1), 2.1 (0.6), and 1.9 (0.4) ng/g, respectively. Brand 3 had quantifiable levels of OTA, with a mean (SD) of 2.3 (0.4) ng/g. Estimated exposure indicated a risk of AFs in all scenarios, and of FUM for brand 1 consumption, OTA and DON for brand 3 consumption, and OTA for brand 4 consumption in the mLB worst-case scenario. In general, mycotoxin levels were below the Chilean regulatory limits, but most of them were above the EU regulation for processed cereal-based food in young children. Because the risk was higher in the 2- to 5-year-old children, we recommend special regulations for this group in Chile.

## 1. Introduction

Cereals are staple foods that are commonly consumed throughout the world as essential sources of energy, minerals, fiber, and vitamins [1]. However, these products are susceptible to infection by various fungi before or after harvest during drying or storage [2]. In general, contamination with *Aspergillus* and *Penicillium* fungi occurs due to high temperature and humidity during food drying and storage, which can produce carcinogenic mycotoxins, such as aflatoxins B1, B2, G1, and G2 (AFs) and ochratoxin A (OTA) [3]. AFs are the most widely studied mycotoxins because of their health effects in humans and animals, such as hepatotoxicity, teratogenicity, immunotoxicity [4], and liver cancer, and are the only mycotoxin in group 1, according to the International Agency for Research on Cancer (IARC) [5]. OTA is a potent renal carcinogen in several animal species [6] and is classified as a possible human carcinogen (group 2B) according to IARC [7,8]. *Fusarium* is a major fungal parasite of grains occurring worldwide and produces a wide range of secondary metabolites. The most known are fumonisins B1, B2, and B3 (FUM), which are possible carcinogens according to IARC [5,7,8]; zearalenone (ZEN), a known phytoestrogen; and type A and B trichothecene, such as T-2/HT2 toxins and deoxynivalenol (DON), respectively, which are acutely cytotoxic and strongly immunosuppressive [9]. 

Because of the health effects of mycotoxins, efforts have been made to reduce human exposure, particularly through food intake. Among those efforts, maximum limits food commodities have been set [10,11]. The limits on mycotoxin levels in Chile are specified in the Food Sanitary Regulation [12] and enforced by the National Mycotoxins Surveillance Program of the Ministry of Health. It started in 2009 with the monitoring of AFs (limit 5 µg/kg), AFM1 (limit 0.05 µg/kg), and ZEN (200 µg/kg), with no specification of the foodstuffs to analyze. In 2013, a new regulation was set, with increased limits of AFs (10 µg/kg) and AFM1 (0.5 µg/kg) and including regulations for OTA (5 µg/kg), patulin (50 µg/kg), FUM (1000 µg/kg), and DON (750 µg/kg). Compared to the regulation of the European Union (EU) [11], the Chilean regulation is less specific since sensitive groups were not considered when determining the limits. For example, in the EU, processed cereal-based foods and baby foods for infants and young children have a special limit of 200 µg/kg for DON and FUM versus 750 and 1000 µg/kg, respectively, in the Chilean regulation. Additionally, the EU regulation includes a more detailed list of foodstuffs with different mycotoxin limits according to the processing and amount consumed.

In Chile, there is little information on human exposure to different mycotoxins although some studies have shown the presence of aflatoxin–lysine adducts in plasma [13] and aflatoxin B1 and M1 in urine [14], with low to medium exposure. Moreover, a high prevalence of OTA at low levels has been reported in urine, plasma, and breast milk [15,16,17]. A current assessment based on urine biomarkers in Chilean adults showed high prevalence and concentrations of DON and low prevalence but high concentrations of ZEN metabolites [14]. According to the Chilean mycotoxin surveillance program, the most commonly contaminated foods are imported spices, with AFs and OTA being the most analyzed mycotoxins [18]. 

Currently, there is no information on mycotoxin exposure in children in Chile. This group has been shown to be the most vulnerable to mycotoxin’s effects [19,20], representing a unique combination of a restricted range of food types, high-calorie intake with respect to body weight, and a decreased ability to eliminate toxins due to immature excretory and metabolic processes [21,22]. In this regard, the present study aimed to assess the exposure to AFs, OTA, ZEN, FUM, DON, and T-2/HT-2 toxins associated with the consumption of breakfast cereals in children aged 2 to 13 years old and to corroborate whether the current Chilean regulation is protecting this group.

## 2. Results

### 2.1. Occurrence and Levels of Mycotoxins in Cereals

No T2/HT2 toxins were detected in the samples. Brands 1 and 2, both 100% corn, had the lowest occurrence of AFs (13 and 40%, respectively), while no samples were positive for OTA. In the case of brands 3 and 4, all samples were over LOD and LOQ values for AFs but below the limit of Chilean regulation (10 ng/g). For OTA, 57% of samples were over LOD in cereal 3, and 100% were over LOD with no quantifiable samples in cereal 4. In general, few samples had quantifiable levels of ZEN, FUM, and DON except for cereal 1, which had three FUM samples and one DON sample over the Chilean regulation (>1000 and >750 ppb, respectively) (Table 1). Levels of each sample and brand are shown in the Appendix A.

Considering mean levels with the modified lower bound (mLB) and upper bound (UB) approach, AFs levels were in the ranges 0.07–0.6 ng/g in brand 1, 0.25–0.72 ng/g in brand 2, 2.15 ng/g in brand 3, and 1.9 ng/g in brand 4. For ZEN, levels were in the ranges 13–25 ng/g in brand 1, 1–8 ng/g in brand 2, 5–24 ng/g in brand 3, and 11–17 ng/g in brand 4. For FUM, levels were in the ranges 304–631 ng/g in brand 1, 0–200 ng/g in brand 2, 140–724 ng/g in brand 3, and 50–360 ng/g in brand 4. For DON, levels were in the ranges 95–392 ng/g in brand 1, 77–393 ng/g in brand 2, 166–518 ng/g in brand 3, and 160–440 ng/g in brand 4. For OTA, levels were in the ranges 0–1 ng/g in brands 1 and 2 and 1–2 ng/g in brands 3 and 4.

Co-occurrences were observed for cereal 1 between ZEN and FUM; ZEN and DON; ZEN, FUM, and DON; and AFs, ZEN, and FUM in 40%, 37%, 27%, and 7% of the samples, respectively. In cereal 2, 20% of the samples had AFs and DON. In cereal 3, all samples had AFs and DON; 83% had AFs, ZEN, and DON; and 53% had AFs, OTA, ZEN, FUM, and DON. In the case of cereal 4, all samples had AFs and OTA, 85% also had DON, 50% had AFs and ZEN, and 15% had all five mycotoxins.

### 2.2. Exposure Assessment

Regarding the estimated daily intakes (EDI) of mycotoxins through cereal consumption, higher EDI was seen in the youngest group, especially FUM for cereal 1 consumption (up to 3262 ng/kg body weight (bw) per day). EDI for each group and scenario is shown in Table 2. Exposure estimations ranged from 1 to 6 ng/kg bw per day for AFs, 0.32 to 4 ng/kg bw per day for OTA, 2 to 45 ng/kg bw per day for ZEN, 19 to 3262 ng/kg bw per day for FUM, and 36 to 2512 ng/kg bw per day for DON (Table 2).

### 2.3. Risk Characterization

The estimated margin of exposure (MOE) values ranged from 13,333 to 70 for AF exposure using the mLB approach. In general, the calculated MOE was below 10,000 with the exception of the mean mLB estimation in the 6-to-13-year-old group due to cereal 1 consumption. The estimation of chronic dietary OTA exposure resulted in levels ranging from 0 to 4.29 ng/kg bw per day. When compared with BMDL10, MOE ranged from 30,851 to 3988, and values were lower than 10,000 in the mLB percentile 95% (P95) of the levels and worst-case scenario (WCS) for cereals 3 and 4 and for all cereals in the UB WCS.

Low risk (i.e., hazard quotient (HQ) < 1) of *Fusarium* mycotoxins ZEN, FUM, and DON was estimated in the mean values in mLB and UB scenarios. However, in P95 and WCS, the HQ was >1 for FUM and DON, indicating concern for children exposed to these mycotoxins through the consumption of cereal 1 and 3 and FUM for cereal 4 consumption (Table 3).

## 3. Discussion

In general, the samples of the four brands of cereals analyzed in the present study had low levels of the studied mycotoxins. The AFs levels in breakfast cereals found in this study ranged from 1.3 to 2.1 ng/g, similar to values reported in Pakistan, ranging from 1.45 to 2.25 ng/g [23], but higher than those shown in Portugal (0.027–0.028 ng/g) [24]. The mean levels of AFs in this study were below the Chilean regulation (10 ng/g) but above the EU regulation for processed cereal-based foods and baby foods for infants and young children (0.1 ng/g) [11]. In this regard, special regulations for children, such as the European regulation, are strongly suggested for foods that are usually consumed by this age group, such as cereals, fruit juices and compotes, and dairy products. This could also be recommended for OTA even though its occurrence and levels in this study were even lower than those of AF; the only quantifiable samples were from cereal 3, a whole-grain cereal with 44% wheat and 28% corn. Levels of OTA found in cereals were similar to values in Pakistan (1.71–2.89 ng/g) [23] and higher than values in Europe (0.026–0.145 ng/g) [24,25,26] and Canada (0.12–0.61 ng/g) [27]. The brand with the highest occurrence of AFs and OTA was cereal 4, a national brand. The highest levels of these mycotoxins were seen in cereal 3, an imported brand.

Interestingly, FUM, DON, and ZEN generally showed low levels in breakfast cereals except for FUM in the gluten-free cornflakes of brand 1, imported from Mexico. The maximum levels of FUM (1970 ng/g) were similar to other studies, with maximum levels of 1980 ng/g in Canada [27] and 2026 ng/g in Portugal [28]. 

In this study, the estimated intakes of FUM showed high concern from a public health point of view in the WCS (HQ > 1), especially for the consumption of brands 1, 3, and 4. Moreover, the maximum EDI of 3261 ng/kg bw per day observed for FUM exceeded by three times the tolerable daily intake (TDI) established of 1000 ng/kg per day but was lower than the estimated intakes seen in some countries, such as Guatemala, Zimbabwe, and China, with maximum EDIs of 7700 ng/kg bw per day for adults [29].

Low levels of ZEN were observed in this study (<84 ng/g), which are even lower than those found in Europe (<172 ng/g) [25,26,30]. In this case, all HQ calculated in the different exposure models for ZEN were <1, indicating no cause of concern for children exposed to ZEN through breakfast cereal consumption.

The levels of DON observed in breakfast cereals (<LOQ–860 ng/g) were higher than those found in Brazil (<LOQ–120.8 ng/g) [31], with maximum levels similar to those found in Canada (940 ng/g) [27] and Belgium (718 and 1295 ng/g) [32,33]. These maximum levels lead to exposures higher than the TDI (HQ > 1) in the 2-to-5-year-old group, indicating health concern for this group through consumption of brands 1 and 3.

Cereals and related food products derived from grains are frequently contaminated with different species of fungi, and co-occurrence of mycotoxins is frequently reported [34,35]. Co-occurrences were observed in all cereals sampled in this study; cereals 3 and 4 showed to have all the mycotoxins studied in some of their batches. The most frequent co-occurrence was ZEN and DON (53% of all samples), followed by AFs and DON (35% of all samples) and AFs and OTA (35% of all samples). In general, most mycotoxin mixtures lead to additive or synergistic effects [35] although current regulations do not consider exposure to multiple mycotoxins. Studying the interactions between multiple mycotoxins in different matrices should be a priority along with incorporating co-contamination in future regulations [36].

Among the limitations of the study were the high LOD and LOQ values of ELISA for FUM, DON, and T2/HT2 toxins. Thus, the occurrence of mycotoxins in cereal may be higher than reported in this study. For example, the LOQ of FUM was the same as the Chilean regulation limits, so all quantifiable samples were over the regulation, and we could not assess the implications of lower levels in the exposure and risk of the children. In the case of T2/HT2 toxins, no official analysis has been made in Chile for these mycotoxins, so levels could be lower than the LOD of this method (10 ng/g), which does not imply necessarily that are not present. In addition, the results may not represent national exposure, as the cereals were sampled in central Chile. Despite these limitations, this is the first large sampling of breakfast cereals in Chile with the aim of determining the levels of the main mycotoxins and exposure to these contaminants in children through consumption.

In conclusion, the levels of mycotoxins, in general, were below the Chilean regulatory limits, but most of the levels found were above the EU regulation for processed cereal-based foods and baby foods for infants and young children. Co-occurrence of mycotoxins in the breakfast cereals showed that children are exposed to multiple mycotoxins, and a cumulative risk assessment is needed. MOE below 10,000 for AFs in almost all the scenarios assessed and the HQ > 1 shown in some of the WCS for FUM, DON, and OTA indicate a cause of concern for children exposed through cereal consumption. Because the risk was higher in the 2-to-5-year-old group, we recommend special regulations for infants and young children in Chile. 

## 4. Methods

### 4.1. Cereal Sampling 

Sampling was based on the UK’s Food Standards Agency [37] and European Commission (EC) Regulation 401/2006 [38] on mycotoxin sampling in cereals on the retail market, which suggests a minimum of *n* = 3 incremental samples and 1 kg of aggregate sample weight. To guarantee representativeness, we sampled the four most commonly consumed breakfast cereals according to the National Food Consumption Survey (ENCA) [39]: brand 1, cornflakes; brand 2, cornflakes; brand 3, chocolate flavor, 44% wheat, and 28% corn; and brand 4, chocolate flavor, 25% wheat, and 25% corn.

For brands 1, 2, and 3, we sampled 30 lots, and for brand 4, we sampled 20 lots, with 3 sub-lots each, from several national supermarkets in central Chile between November 2019 and June 2021. We sampled a total of 330 samples and 110 lots. Each batch was milled in a Romer mill (Romer Seris II Mill, Union, MO, USA) to increase homogeneity and representativeness. The grounded samples were collected in sterile plastic bags and stored at −20 °C until further analysis.

### 4.2. Analysis of Cereal Samples 

The cereal samples were analyzed by enzyme-linked immunoassay (ELISA) using Veratox^®^ HS (Neogen) for total AFs and Veratox^®^ for OTA; total FUM, ZEN, and DON (5/5, regarding the time of incubation of the method); and T-2/HT-2 (Neogen Inc., Lansing, MI, USA). The method is based on the antibody–antigen interaction and approved by the AOAC Research Institute for FUM (AOAC 2001.06) and by the Federal Grain Inspection (FGIS) of the USDA for DON (FGIS 2018-110).

The extraction and quantification of the mycotoxins in cereal were carried out according to the manufacturer’s instructions. Briefly, 10 g of the grounded cereal was weighed into a special cup with a lid and mixed with 50 mL of 70% methanol in the case of AFs, ZEN, and FUM; 40 mL of 50% methanol in the case of OTA and T-2/HT-2; and 100 mL of distilled water in the case of DON. Subsequently, the mixture was shaken for 3 min at 300 rpm (N-Biotek, Si GyeongGi-DO, China) and filtered (Whatman No. 1 paper). In the case of DON, the mixture was centrifugated (Eppendorf Centrifuge Model 5804 R, Hamburg, Germany) at 5000 rpm for 10 min after shaking, and the supernatant was kept for analysis. Later, 100 µL of each extract was used for the ELISA procedure. Spectrometric analysis was performed using a microplate reader (PHomo Autobio Microplate Reader, Zhengzhou, Henan, China) at 650 nm. The measured absorbance was automatically converted to the mycotoxin concentration unit ng/g. The limit of detection (LOD) and limit of quantification (LOQ) were assumed according to the manufacturer’s guidelines: 0.5 and 1.0 ng/g for AFs; 1 and 2 ng/g for OTA; 5 and 25 ng/g for ZEN; 200 and 1000 ng/g for FUM; 100 and 500 ng/g for DON; and 10 and 25 ng/g for T2/HT2 toxin, respectively. 

The verification of the method was carried out by analyzing quality control material (Trilogy Analytical Laboratory, Washington, DC, USA) of corn naturally contaminated with multiple mycotoxins (AFs, OTA, ZEN, FUM, DON, T2/HT2 toxin). Recovery ranged from 97.9 to 100.1% and the variation coefficient from 5.1 to 15.6%. 

### 4.3. Exposure Estimation

The estimated daily intake (EDI) of mycotoxins for children was estimated according to Equation (1) [40]
EDI = mycotoxin levels (ng/g) × consumption (g/day)/body weight (kg)(1)

Using mycotoxin levels (ng/g) found in cereals with a modified lower bound (mLB) and upper bound (UB) approach according to EFSA [41] and mean consumption by age group (2–5 years and 6–13 years) according to ENCA [39] and adjusted by the mean weight of each group according to the Health Ministry of Chile and the World Health Organization (WHO) [42,43]. In addition, a worst-case scenario (WCS) was estimated with higher mycotoxin levels, maximum consumption, and lower body weight of the group. Parameters used in the estimations are summarized in Table 4.

### 4.4. Risk Characterization

The risk characterization for noncarcinogenic mycotoxins was evaluated with a hazard quotient (HQ) value between the EDI estimates and the international information available on tolerable daily intake (TDI) according to EFSA (Equation (2)):HQ = EDI/TDI(2)

A TDI of 1.0 μg/kg bw per day was used for FUM [44], 0.25 μg/kg bw per day for ZEN [45], and 1 μg/kg bw per day for DON [46]. HQ > 1 was considered to indicate risk. For AFs and OTA, risk was estimated by the margin of exposure (MOE) according to Equation (3), considering a benchmark dose lower confidence limit for a response of 10% (BMDL10) of 0.4 μg/kg bw per day for the incidence of HCC in male rats following aflatoxin B1 exposure [47] and a BMDL10 of 14.5 μg/kg bw per day for OTA calculated from kidney tumors seen in rats [48]. MOE < 10,000 was considered to indicate high health concern.
MOE = BMDL10/EDI(3)

## Figures and Tables

**Table 1 toxins-14-00324-t001:** Occurrence and levels of aflatoxins (AFs), ochratoxin A (OTA), zearalenone (ZEN), fumonisins (FUM), and deoxynivalenol (DON) in breakfast cereals.

		AFs	OTA	ZEN	FUM	DON
Brand 1 Cornflakes	>LOD (%)	13	0	67	50	73
	>LOQ (%)	0	0	20	17	3
	Mean (SD) level (ng/g)	<LOQ	<LOQ	54 (20)	1552 (351)	706 (218)
	Max. Level (ng/g)	<LOQ	<LOQ	84	1970	860
Brand 2 Cornflakes	>LOD (%)	40	0	13	0	80
	>LOQ (%)	10	0	0	0	0
	Mean (SD) level (ng/g)	1.3 (0.1)	<LOQ	<LOQ	<LOQ	<LOQ
	Max. level (ng/g)	1.4	<LOQ	<LOQ	<LOQ	<LOQ
Brand 3: 44% wheat/28%corn	>LOD (%)	100	57	97	67	100
	>LOQ (%)	100	10	0	0	13
	Mean (SD) level (ng/g)	2.1 (0.6)	2 (0.4)	<LOQ	<LOQ	630
	Max. level (ng/g)	3.1	2.7	<LOQ	<LOQ	700
Brand 4: 25% wheat/25%corn	>LOD (%)	100	100	60	25	85
	>LOQ (%)	100	0	0	0	0
	Mean (SD) level (ng/g)	1.9 (0.4)	<LOQ	<LOQ	<LOQ	<LOQ
	Max. level (ng/g)	2.7	<LOQ	<LOQ	<LOQ	<LOQ

SD, standard deviation; LOD, limit of detection; LOQ, limit of quantification.

**Table 2 toxins-14-00324-t002:** Estimated daily intake (EDI) in ng/kg bw per day of aflatoxins (AFs), ochratoxin A (OTA), zearalenone (ZEN), fumonisins (FUM), and deoxynivalenol (DON) through consumption of four breakfast cereals by children 2–5 years old and 6–13 years old.

**EDI (ng/kg bw/day)**	**2–5 Years**	**mLB-UB Mean**		**mLB-UB P95**		**mLB-UB WCS**	
	AF	0.06	0.49	0.42	0.83	0.91	1.82
	OTA	0.00	0.83	0.00	4.91	0.00	1.82
	ZEN	10.64	20.78	60.48	60.48	131.95	132.73
	FUM	253.33	525.56	1495.00	1495.00	3261.82	3261.82
	DON	79.44	326.67	368.33	551.67	803.64	1203.64
Brand 1	**6–13 Years**	**mLB-UB Mean**		**mLB-UB P95**		**mLB-UB WCS**	
	AF	0.03	0.27	0.24	0.47	1.40	1.40
	OTA	0.00	0.47	0.00	2.11	1.40	1.40
	ZEN	6.00	11.72	34.11	34.11	102.20	102.20
	FUM	142.88	296.41	843.18	843.18	2511.60	2511.60
	DON	44.81	184.24	207.74	311.14	926.80	926.80
	**2–5 Years**	**mLB-UB Mean**		**mLB-UB P95**		**mLB-UB WCS**	
	AF	0.21	0.60	1.08	1.08	2.36	2.36
	OTA	0.00	0.83	0.00	0.83	0.00	1.82
	ZEN	0.67	6.92	4.17	20.83	9.09	45.45
	FUM	0.00	166.67	0.00	166.67	0.00	363.64
	DON	63.89	327.78	83.33	416.67	181.82	909.09
Brand 2	**6–13 Years**	**mLB-UB Mean**		**mLB-UB P95**		**mLB-UB WCS**	
	AF	0.12	0.34	0.73	0.73	1.82	1.82
	OTA	0.00	0.47	0.00	0.56	0.00	1.40
	ZEN	0.38	3.90	2.80	14.00	7.00	35.00
	FUM	0.00	94.00	0.00	112.00	0.00	280.00
	DON	36.03	184.87	56.00	280.00	140.00	700.00
	**2–5 Years**	**mLB-UB Mean**		**mLB-UB P95**		**mLB-UB WCS**	
	AF	1.78	1.78	2.61	2.61	5.69	5.69
	OTA	058	1.34	1.97	1.97	4.29	4.29
	ZEN	4.03	20.26	4.17	20.83	9.09	45.45
	FUM	116.67	603.45	166.67	833.33	363.64	1818.18
	DON	138.33	431.61	574.17	574.17	1252.73	1252.73
Brand 3	**6–13 Years**	**mLB-UB Mean**		**mLB-UB P95**		**mLB-UB WCS**	
	AF	1.01	1.01	1.47	1.75	4.38	4.38
	OTA	0.32	0.76	1.11	1.32	3.30	3.30
	ZEN	2.27	11.43	2.35	14.00	7.00	35.00
	FUM	65.80	340.35	94.00	560.00	280.00	1400.00
	DON	78.02	243.43	323.83	385.84	964.60	964.60
	**2–5 Years**	**mLB-UB Mean**		**mLB-UB P95**		**mLB-UB WCS**	
	AF	1.58	1.58	2.22	2.22	4.84	4.84
	OTA	0.83	1.67	0.83	1.67	1.82	3.64
	ZEN	9.08	14.17	17.84	20.83	38.93	45.45
	FUM	41.67	300.00	166.67	833.33	363.64	1818.18
	DON	132.92	366.67	239.17	416.67	521.82	909.09
Brand 4	**6–13 Years**	**mLB-UB Mean**		**mLB-UB P95**		**mLB-UB WCS**	
	AF	0.71	0.89	1.25	1.25	3.72	3.72
	OTA	0.38	0.94	0.47	0.94	1.40	2.80
	ZEN	4.09	7.99	10.06	11.75	29.97	35.00
	FUM	18.80	169.20	94.00	470.00	280.00	1400.00
	DON	59.96	206.80	134.89	235.00	401.80	700.00

mLB, modified lower bound (levels were assumed <Limit of detection LOD = 0 and <Limit of quantification LOQ = LOD); UB, upper bound (<LOD = LOD of the method and <LOQ = LOQ of the method); WCS, worst-case scenario (higher mycotoxin levels, maximum consumption, and lower body weight of the group); P95, percentile 95% of the levels. The LOD and LOQ were assumed according to the manufacturer’s guidelines: 0.5 and 1.0 ng/g for AFs; 1 and 2 ng/g for OTA; 5 and 25 ng/g for ZEN; 200 and 1000 ng/g for FUM; 100 and 500 ng/g for DON; and 10 and 25 ng/g for T2/HT2 toxin, respectively.

**Table 3 toxins-14-00324-t003:** Summary of the risk of mycotoxins (aflatoxins (AFs), ochratoxin A (OTA), zearalenone (ZEN), fumonisins (FUM), and deoxynivalenol (DON)), in children 2–5 years old and 6–13 years old because of the consumption of breakfast cereal.

Groups	2–5 years	6–13 years	2–5 years	6–13 years
	mLB mean values		mLB WCS	
Brand 1	AFs	-	AFs, FUM	AFs, FUM
Brand 2	AFs	AFs	AFs	AFs
Brand 3	AFs	AFs	AFs, OTA, DON	AFs, OTA
Brand 4	AFs	AFs	AFs, OTA	AFs
	UB mean values		UB WCS	
Brand 1	AFs	AFs	AFs, OTA, FUM, DON	AFs, FUM
Brand 2	AFs	AFs	AFs, OTA	AFs
Brand 3	AFs	AFs	AFs, OTA, FUM, DON	AFs, OTA, FUM
Brand 4	AFs, OTA	AFs	AFs, OTA, FUM	AFs, OTA, FUM

mLB, modified lower bound (levels were assumed <Limit of detection LOD = 0 and <Limit of quantification LOQ = LOD); UB, upper bound (<LOD = LOD of the method and <LOQ = LOQ of the method); WCS, worst-case scenario (higher mycotoxin levels, maximum consumption, and lower body weight of the group).

**Table 4 toxins-14-00324-t004:** Parameters used to estimate children’s exposure to mycotoxin through cereal consumption.

Groups	Minimum	Mean	Maximum
**2–5 years old**			
Weight (kg) ^a^	11	15	19
Consumption (g) ^b^	7	13	20
**6–13 years old**			
Weight (kg) ^a^	20	30	50
Consumption (g) ^b^	6	14	28

^a^ Body weight of children according to the Health Ministry of Chile and the World Health Organization (WHO) [42,43]. ^b^ According to the National Food Consumption Survey (ENCA) [39].

## Data Availability

The original data is provided in Appendix A.

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
