# Peer review of "Mycotoxin Exposure in Children through Breakfast Cereal Consumption in Chile"

_toxins, 2022, doi:10.3390/toxins14050324_

Round 1

Reviewer 1 Report

In this manuscript, the authors determined the level of mycotoxin contamination of breakfast cereals in Chile and estimated the risk of exposure for children.

The subject is well introduced with regard to the existing legislation on the mycotoxin content of foods. Nevertheless, information on the recommended daily exposure limit per kg for children should be added to compare the authors' results with the recommendations.

The results are detailed and well presented. Nevertheless, there is some ambiguity in the introduction of the different blocks of results. In particular, Lines 111-115. I suggest to replace “Regarding exposure” with for example : “Regarding the level of mycotoxins intake by children”

The methods need to be improved. In particular Lines 210-216. Please clarify the different steps of samples processing described in the lines 210-216. Did the authors really used methanol solution directly for ELISA?

Discussion

The first paragraph need to be more precise on the data compared and the structure of the sentences need to be improved. It needs to start with a summary of the major findings of the study.

Line 137-138 – The authors need to mention that it concerns the findings of the present manuscript in Chile.

For example: The AF levels in breakfast cereals has been found to be lower or similar in Chile than in Pakistan…..

In general the usage of the expression “may pose a risk” is not sufficiently clear. Please clarify in each context to which data you are refering : tolerable daily intake per kg ? or margin of exposure.

Line 139-140: The authors need to be more precise. Is ”this carcinogenic mycotoxin […] considered to pose a risk in almost all scenarios” of their study or of the three studies cited in the previous sentence.

Line 163-165. Please reformulate and develop why these levels may pose a high risk to the youngest group of consumers through consumption of brands 1 and 3.

Concerning the limitations, Lines 175-176 the authors mentioned that the LOD and LOQ values of the technic used are high. They have to develop and compare if these values are sufficient to detect the most of risk samples (values by comparison to the recommended concentrations)

In Conclusion, the authors concluded on the level of mycotoxins, they need to specifiy that this concerns breakfast cereals. Please clarify in the context, what do you mean by most of them were above the EU regulation for small children

Author Response

Dear reviewer,

We highly appreciate the overall positive feedback and the constructive comments of the reviewers regarding our manuscript “Mycotoxin exposure in children through breakfast cereal consumption in Chile”. The reviewers’ comments were carefully considered in the revised version. A detailed response to their comments is provided in this letter.

We hope that you will find the revised version of the manuscript now acceptable for publication in Toxins and look forward to your response.

With best regards,

In this manuscript, the authors determined the level of mycotoxin contamination of breakfast cereals in Chile and estimated the risk of exposure for children.

The subject is well introduced with regard to the existing legislation on the mycotoxin content of foods. Nevertheless, information on the recommended daily exposure limit per kg for children should be added to compare the authors' results with the recommendations.

R: Thank you for the suggestion. Because each of the mycotoxins has different tolerable daily intakes, and even carcinogenic (aflatoxins and ochratoxin A) and no- carcinogenic mycotoxins have a different way to assessed the risk (with a MOE or HQ respectively), we found it more clear putting that information in materials and methods, along with the formulas used (point 4.4).

The results are detailed and well presented. Nevertheless, there is some ambiguity in the introduction of the different blocks of results. In particular, Lines 111-115. I suggest to replace “Regarding exposure” with for example : “Regarding the level of mycotoxins intake by children”

R: Was changed as suggested

The methods need to be improved. In particular Lines 210-216. Please clarify the different steps of samples processing described in the lines 210-216. Did the authors really used methanol solution directly for ELISA?

R: The quantification of mycotoxins consists of two parts, first the preparation of the sample and extraction, and second, the quantification process. In the first part, the cereal is grounded and extracted mainly with methanol. Depending on the mycotoxin is the solvent to use. For the second part, the manufacturer provides everything necessary to carry out the quantification. The two parts are carried out according to the instructions of the kit supplier. The phrase “The extraction and quantification of the mycotoxins in cereal were carried out according to the manufacturer’s instructions. Briefly …” was added. The complete protocols can be found at: https://www.neogen.com/en-gb/categories/mycotoxins/

Discussion

The first paragraph need to be more precise on the data compared and the structure of the sentences need to be improved. It needs to start with a summary of the major findings of the study.

R: The phrase “In general, the samples of the 4 brands of cereals analyzed in the present study, had low levels of the studied mycotoxins“  was added.

Line 137-138 – The authors need to mention that it concerns the findings of the present manuscript in Chile.For example: The AF levels in breakfast cereals has been found to be lower or similar in Chile than in Pakistan…..

R: The sentence was re-phrased to “The AFs levels in breakfast cereals found in this study ranged from 1.3 to 2.1 ng/g, similar to values reported in Pakistan, ranging from 1.45 to 2.25 ng/g [23], but higher than those shown in Portugal (0.027–0.028 ng/g) “

In general the usage of the expression “may pose a risk” is not sufficiently clear. Please clarify in each context to which data you are refering : tolerable daily intake per kg ? or margin of exposure.

R: Thank you for the suggestion. The following sentences were re-phased:

Line 160: “In this study, the EDI of FUM for the consumption of brands 1, 3, and 4, showed high concern from a public health point of view in the WCS (HQ > 1).

L168: “The levels of DON observed in breakfast cereals (<LOQ - 860 ng/g) were higher than those found in Brazil (<LOQ - 120.8 ng/g) [31], with maximum levels similar to those found in Canada (940 ng/g) [27] and Belgium (718 and 1295 ng/g) [32, 33]. These maximum levels lead to exposures higher than the TDI in the 2 to 5 years old group, resulting in health concern through consumption of brands 1 and 3”.

Line 139-140: The authors need to be more precise. Is ”this carcinogenic mycotoxin […] considered to pose a risk in almost all scenarios” of their study or of the three studies cited in the previous sentence.

R: The sentence was re-phrased to “The mean levels of AFs in this study were below the Chilean regulation (10 ng/g) but above the EU regulation for processed cereal-based foods and baby foods for infants and young children (0.1 ng/g) [11]. In this regard, special regulations for children, such as the European regulation, are strongly suggested for foods that are usually consumed by this age group, such as cereals, fruit juices and compotes, and dairy products”.

Line 163-165. Please reformulate and develop why these levels may pose a high risk to the youngest group of consumers through consumption of brands 1 and 3.

R: The sentence was re-phrased to “These maximum levels lead to exposures higher than the TDI in the 2 to 5 years old group, resulting in health concern through consumption of brands 1 and 3”.

Concerning the limitations, Lines 175-176 the authors mentioned that the LOD and LOQ values of the technic used are high. They have to develop and compare if these values are sufficient to detect the most of risk samples (values by comparison to the recommended concentrations)

R: The sentence was re-phrased to “Among the limitations of the study were the high LOD and LOQ values of ELISA for FUM, DON, and T2/HT2 toxins. Thus, the occurrence of mycotoxins in cereal may be higher than reported in this study. For example, the LOQ of FUM was the same as the Chilean regulation limits, so all quantifiable samples were over the regulation and we couldn´t assess the implications of lower levels in the exposure and risk of the children. In the case of T2/HT2 toxins, no official analysis has been made in Chile for these mycotoxins, and levels could be lower than the LOD of this method (10 ng/g), which doesn’t imply necessarily that is not present”

In Conclusion, the authors concluded on the level of mycotoxins, they need to specifiy that this concerns breakfast cereals. Please clarify in the context, what do you mean by most of them were above the EU regulation for small children

R: The sentence was re-phrased to “In conclusion, the levels of mycotoxins, in general, were below the Chilean regulatory limits, but most of the levels found were above the EU regulation for processed cereal-based foods and baby foods for infants and young children. These differences in regulatory limits could explain the resulting health concern of AFs in almost all the scenarios assessed, and FUM, DON, and OTA in the worst-case scenario. Because the risk was higher in the 2-to-5-year-old group, we recommend special regulations for infants and small children in Chile.”. We hope this way is clearer.

Reviewer 2 Report

Dear Authors,

I am pleased to have opportunity to review your manuscript describing the survey of mycotoxin occurrence in cereals in Chile and the exposure of children through breakfast cereal consumption. The manuscript is interesting, well structured, and clear, and it can be read well. Publication is recommended after corrections according to the comments bellow.

Lines 13 and 233: I suppose “(UP)” should be “(UB)” as at some other places in the text.

Line 33: I suggest changing “on the farm” to e.g., “in the field” or “before harvest”. To my opinion “on the farm” can be before or after harvest.

Line 51: Is the regulation [12] from the year 1997 still in force/applicable? You mentioned in line 54, that in 2013 a new regulation was set. I suggest adding it to the references and refer to it (instead of [12] if it is not relevant anymore.

Lines 70–71: I suggest changing “with AFs and OTA, the most sampled and analyzed mycotoxins” to “with AFs and OTA, the most analyzed mycotoxins”.

Table 2: I suggest adding an explanation what UB mean, UB P95 and UB WCS are.

Line 193: I suggest changing “the 4 most commonly” to “the four most commonly.

Line 205: Please explain what 5/5 means.

Line 217: I suggest changing “with an optic density of 650 nm” to “at 650 nm”.

Line 272: Please check all references carefully in order to correct inconsistencies in the style, author order, names e.g., [6] (to accomplish the author names), [9] (to correct the names and author order), [14] (to use family names instead of first names), [26] (to use initials instead of first names), [28, 32] (to use initials after names) …

Line 360: “2006, 24, 1–42” shall be “2006, 70, 12–34”.

Line 366: Please check/correct the author names.

Author Response

Dear reviewer,

We highly appreciate the overall positive feedback and the constructive comments of the reviewers regarding our manuscript “Mycotoxin exposure in children through breakfast cereal consumption in Chile”. The reviewers’ comments were carefully considered in the revised version. A detailed response to their comments is provided in this letter.

We hope that you will find the revised version of the manuscript now acceptable for publication in Toxins and look forward to your response.

With best regards,

Reviewer 2

I am pleased to have opportunity to review your manuscript describing the survey of mycotoxin occurrence in cereals in Chile and the exposure of children through breakfast cereal consumption. The manuscript is interesting, well structured, and clear, and it can be read well. Publication is recommended after corrections according to the comments bellow.

Thank you very much for your kind words and useful suggestions for improvement.

Lines 13 and 233: I suppose “(UP)” should be “(UB)” as at some other places in the text.

R: Yes, indeed. It was replaced.

Line 33: I suggest changing “on the farm” to e.g., “in the field” or “before harvest”. To my opinion “on the farm” can be before or after harvest.

R: Done

Line 51: Is the regulation [12] from the year 1997 still in force/applicable? You mentioned in line 54, that in 2013 a new regulation was set. I suggest adding it to the references and refer to it (instead of [12] if it is not relevant anymore.

R: The Chilean Food Sanitary has been updated since its creation. The latest update was on February 2021; that date was added to references.

Lines 70–71: I suggest changing “with AFs and OTA, the most sampled and analyzed mycotoxins” to “with AFs and OTA, the most analyzed mycotoxins”.

R: Done

Table 2: I suggest adding an explanation what UB mean, UB P95 and UB WCS are.

R: Done

Line 193: I suggest changing “the 4 most commonly” to “the four most commonly.

R: Done

Line 205: Please explain what 5/5 means.

R: Is the name of the ELISA. The brand has 3 kinds of ELISA for DON. DON 5/5 refers to the incubation time. The protocol involves 2 incubations and each of them is 5 minutes. There are other formats that use 2 and 3 minutes of incubation and are called DON 2/3. There is also the DON HS (high sensitive). The clarification “DON (5/5 regarding the time of incubation of the method)” was added.

Line 217: I suggest changing “with an optic density of 650 nm” to “at 650 nm”.

R: Done

Line 272: Please check all references carefully in order to correct inconsistencies in the style, author order, names e.g., [6] (to accomplish the author names), [9] (to correct the names and author order), [14] (to use family names instead of first names), [26] (to use initials instead of first names), [28, 32] (to use initials after names) …

R: Thank you for the revision, the references were checked and modified as suggested

Line 360: “2006, 24, 1–42” shall be “2006, 70, 12–34”.

R: Done

Line 366: Please check/correct the author names.

R: Done

Round 2

Reviewer 1 Report

In the discussion section, the distinction between the mycotoxins level in breakfast cereals and the predicted level of exposure for the target age groups is still not clear enough. 

For example, lines 163-164, the authors added a sentence that seems to contradict the previous sentences on the level of contamination of breakfast cereals by FUM. Please introduce way *the EDI of FUM for the consumption of brands 1, 3, and 4, showed high concern from a public health point of view in the WCS"

The distinction is not clear in lines 168-170 neither.

Please go through the other paragraphs and consider also this distinction.

In the material and methods, the methods of samples preparation before ELISA is still not detailed 

Author Response

Dear Reviewer,

Thanks for your feedback and positive comments.

For clarifying the results, 3 sub items were added in this item. Unfortunately, in discussion all the parts (levels, exposure and risk assessment) are displayed in a joint way and discussed by each mycotoxin. The fact that in this journal methods are in the end, makes more complicated to explain some differences and to be aware of the abbreviations. A full revision of the discussion was made in this sense. I hope this new version in clearer.

Also, details of the samples preparation before ELISA were added. Please, let me know if you need further information.

With kind regards,

[Authors' name - hidden]